# LLaMA32-Med: Parameter-Efficient Adaptation of Multimodal LLMs for Medical Visual Question Answering

**Wanqi Dong**[1]                      WANQI.DONG@U.NUS.EDU
**Jingze Ge**[1]                        E1351377@U.NUS.EDU
**Wanyue Dong**[2]                DONGWANYUE6699@STU.JNU.EDU.CN
**Mehul Motani**[1]                      MOTANI@NUS.EDU.SG
[1] *National University of Singapore*
[2] *Jinan University*

**Editors:** Accepted for publication at MIDL 2026

## Abstract

Artificial intelligence has shown great promise in healthcare, particularly in diagnostic support. While healthcare data is inherently multimodal, existing models struggle to fully leverage diverse clinical data, e.g., images and text. Although recent Multimodal Large Language Models (MLLMs) exhibit strong potential, their performance in medical scenarios is constrained by training on general-domain data and the high computational cost of full-parameter adaptation. In this work, we present a two-stage lightweight adaptation framework for fine-tuning general-purpose MLLMs on medical multimodal tasks. Building on the LLaMA 3.2 Vision-Instruct model, we adopt parameter-efficient fine-tuning techniques that update less than 2% of the model parameters. This enables the injection of domain-specific medical knowledge while requiring approximately 20 GB of GPU memory. Furthermore, we design task-specific and role-based prompting strategies to better guide medical visual understanding tasks. Experimental results show that our approach achieves performance comparable to or surpassing state-of-the-art methods while significantly outperforming the original general-domain model. Comparative evaluations with recent MLLMs highlight the strong adaptability of the LLaMA 3.2 Vision-Instruct backbone, validating its effectiveness as a foundation for practical multimodal medical AI systems.

**Keywords:** Multimodal Large Language Models (MLLMs), Parameter-Efficient Fine-Tuning (PEFT), Medical Visual Question Answering (VQA), Clinical Applications

## 1. Introduction

With the rapid advancement of artificial intelligence (AI) in the medical domain, AI has shown strong potential in tasks such as diagnostic assistance and disease prediction. Applications including medical image segmentation and medical question answering have greatly contributed to the intelligent transformation of healthcare. However, most existing methods still rely on unimodal data processing, while real-world clinical scenarios inherently involve multimodal information such as medical images and textual information. Enabling models to jointly understand and integrate these modalities for complex diagnostic reasoning remains a key challenge in developing practical medical AI systems.

The emergence of Multimodal Large Language Models (MLLMs) offers new opportunities to address this challenge. By combining visual encoders (e.g., ViT (Dosovitskiy et al., 2020)) with language generation models (e.g., LLaMA (Touvron et al., 2023), GPT),

MLLMs can jointly model medical image–text data (Qin et al., 2022) and have demonstrated promise in tasks such as medical image captioning and visual question answering (VQA). In regions with limited medical resources, MLLMs could support diagnostic workflows or serve as interactive clinical assistants. However, most publicly available MLLMs are trained on general-domain corpora and lack medical-specific adaptation. Their visual modules may overlook subtle pathological cues, while their language modules can hallucinate or misinterpret clinical terminology. In addition, the private nature of medical data necessitates efficient and resource-friendly adaptation methods that can be deployed under strict privacy constraints in real clinical senarios.

To tackle these challenges, we propose LLaMA32-Med, an efficient two-stage adaptation framework for transferring general-purpose MLLMs to medical VQA tasks. Using LLaMA 3.2 Vision-Instruct as the backbone, we employ Parameter-Efficient Fine-Tuning (PEFT) to perform lightweight tuning and inject domain-specific medical knowledge. Unlike models such as LLaVA-Med (Li et al., 2023a), which rely on large-scale training and high-end infrastructure, our approach updates only a small fraction of model parameters, significantly reducing computational cost while preserving strong performance. We further design task-specific prompt templates to guide the model in generating clinically meaningful answers.

We evaluate our framework on multiple medical multimodal datasets and extend it to other general-domain MLLMs, including LLaVA (Liu et al., 2023), Qwen2.5-VL (Bai et al., 2025), and Gemma3 (Team et al., 2025) for comparative analysis. Experimental results show that LLaMA32-Med achieves substantial gains over the original general-domain model and delivers competitive or superior performance compared to state-of-the-art medical MLLMs.

**Main Contributions:**
1. We introduce a lightweight two-stage PEFT pipeline for adapting the LLaMA 3.2 Vision-Instruct model, updating less than 2% of parameters and enabling efficient training with only approximately 20 GB of GPU memory;
2. We design task-specific and role-based prompt templates for Medical VQA and medical image description tasks, improving clinical relevance and response accuracy;
3. Extensive experiments on standard medical multimodal benchmarks show that our method matches or surpasses SOTA models despite its minimal parameter update;
4. Comparative studies across multiple general-domain MLLMs highlight the strong adaptability of the LLaMA 3.2 architecture, establishing it as an effective backbone for medical multimodal AI systems.

## 2. Related Work

### 2.1. Medical Large Language Models (LLMs)

Recent progress in medical LLMs has been largely driven by advances in vision–language alignment and instruction tuning, enabling models to handle increasingly complex clinical tasks. LLaVA-Med (Li et al., 2023a) is an early representative that aligns vision encoders using over one million PubMed Central image–caption pairs and is later instruction-tuned on GPT-4 generated data. Although effective, LLaVA-Med updates nearly all parameters, making adaptation and deployment infeasible in resource-constrained environments. In contrast, our method integrates lightweight LoRA adapters into the LLaMA-3.2 Vision-

Instruct backbone, updating only a small fraction of parameters and therefore substantially reducing GPU memory usage and training time while maintaining strong performance.

MedPaLM (Singhal et al., 2023) achieves expert-level results across a wide range of medical tasks but remains closed-source, limiting reproducibility and community-driven development. HuatuoGPT (Zhang et al., 2023), trained with ChatGPT-distilled dialogues and optimized via RLAIF (Lee et al., 2023), is restricted to text-only interactions and thus cannot process medical images. Our model naturally supports both image and text modalities, enabling end-to-end multimodal inference.

## 2.2. Medical Visual Question Answering (Medical VQA)

Medical VQA requires jointly interpreting visual and textual information to identify pathological cues and produce clinically meaningful answers. Traditional methods typically rely on vision encoders with task-specific classification heads. For example, M2I2 (Li et al., 2023b) leverages masked image–language modeling and contrastive alignment to achieve strong performance, but its dependence on specialized heads complicates deployment. Pub-MedCLIP (Eslami et al., 2021) fine-tunes CLIP on PubMed image–text pairs to improve VQA accuracy, yet its loose integration with language decoders limits end-to-end reasoning.

MLLM-based approaches provide deeper multimodal coupling. BiomedGPT (Zhang et al., 2024) unifies a ViT encoder and a BART-style decoder into a coherent sequence-to-sequence pipeline and achieves competitive results through multitask biomedical training. Med-MoE (Jiang et al., 2024) activates expert subnetworks through a trainable router, reaching state-of-the-art accuracy while activating a subset of parameters, but requires careful hyperparameter tuning for optimal routing.

Our approach employs the LLaMA 3.2 Vision-Instruct model, which natively supports image–text dialogue. Through PEFT, we fine-tune the model on relatively small yet diverse medical datasets while updating only a minimal set of parameters, significantly improving adaptability, scalability, and deployment efficiency.

## 3. Methodology

Our method comprises two sequential stages: biomedical alignment followed by downstream fine-tuning. Both stages employ parameter-efficient fine-tuning (PEFT) strategies to systematically optimize the general-domain MLLM. This multi-stage design improves training efficiency, maintains strong task performance, and significantly reduces the overall computational cost required for large-scale medical multimodal adaptation.

In the biomedical alignment stage, we perform instruction tuning on the LLaMA 3.2 Vision-Instruct 11B model to enhance visual understanding of medical images. This stage enables the model to generate accurate, context-aware descriptions in response to medical instructions. Subsequently, during the downstream fine-tuning stage, we leverage the pretrained weights from the first stage to further adapt the model for Medical VQA tasks.

Throughout both stages, we design and apply task-specific prompts tailored to each training objective, ensuring close alignment with the target tasks. The following sections provide a detailed overview of the datasets, model architecture, fine-tuning procedure, and the complete training pipeline.

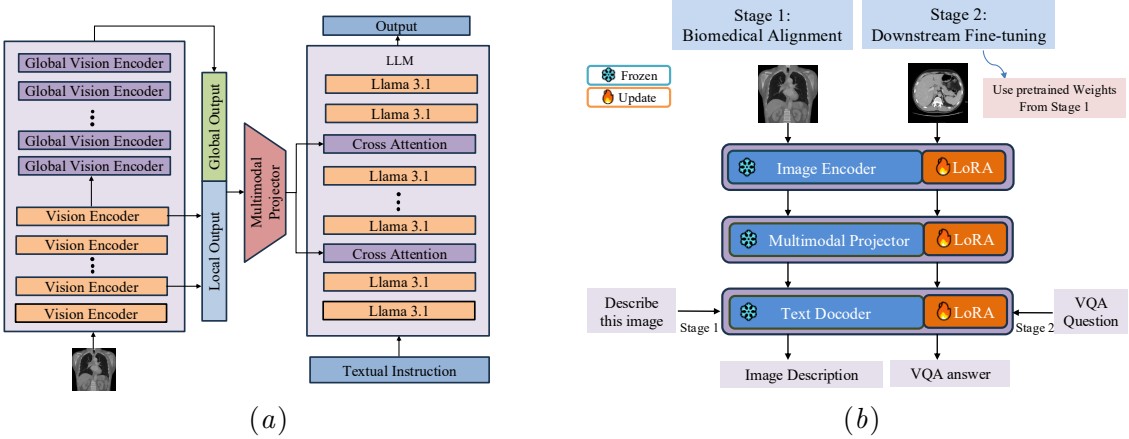

Figure 1: (a) The overall architecture of the foundational model serving as the base for our LLaMA32-Med; (b) Our two-stage fine-tuning pipeline.

### 3.1. Model Structure

In both stages of our framework, we build upon Meta's open-source LLaMA 3.2 Vision-Instruct model (11B) (Meta AI, 2024), a state-of-the-art MLLM that serves as the backbone for multimodal modeling and fine-tuning on medical image–text tasks. The overall architecture is illustrated in Figure 1(a). Compared to commonly used MLLMs such as LLaVA(Liu et al., 2023) and Qwen2-VL(Wang et al., 2024), LLaMA 3.2 Vision-Instruct offers improved capabilities in vision–language alignment and cross-modal integration.

The model comprises three main components: a vision encoder, a multimodal projector, and a language model. A detailed description of the model architecture is provided in Appendix A. Unlike many MLLMs that rely on early fusion strategies by directly concatenating vision and text tokens, LLaMA 3.2 Vision-Instruct adopts a multi-stage cross-attention fusion to enable more effective multimodal interaction. Compared to architectures like LLaVA (Liu et al., 2023) and Qwen2-VL (Wang et al., 2024) that rely on straightforward concatenation-based strategy, LLaMA 3.2 Vision-Instruct demonstrates superior performance in visual semantic modeling, cross-modal reasoning, and generation. This architecture provides a robust foundation for the superior adaptability of our method and makes it a promising backbone for future medical multimodal adaptation.

### 3.2. Fine-Tuning Strategy

Full-parameter fine-tuning of MLLMs often incurs substantial computational costs, high memory usage, and long training time, posing challenges for practical adaptation and deployment. Unlike many state-of-the-art approaches in the medical MLLM domain such as LLaVA-Med, we adopt Parameter-Efficient Fine-Tuning (PEFT) strategies LoRA (Hu et al., 2022) and QLoRA (Dettmers et al., 2023) in two training stages of our framework. Specifically, we insert LoRA adapters into every linear layer of the whole model, and update only these adapters while keeping the rest of the model parameters frozen.

LoRA decomposes the full-rank weight update matrix $\Delta W \in \mathbb{R}^{m \times n}$ into two smaller low-rank matrices $A \in \mathbb{R}^{m \times r}$ and $B \in \mathbb{R}^{r \times n}$, where the rank $r \ll \min(m, n)$. During

training, only $A$ and $B$ are updated, while the original weights $W$ remain frozen. QLoRA further improves memory efficiency by applying 4-bit NormalFloat (NF4) quantization to the frozen pretrained weights, thereby reducing GPU memory usage and mitigating out-of-memory (OOM) issues.

By leveraging LoRA and QLoRA, we fine-tune only 1.24% (134M) of the parameters, with the rest frozen, allowing us to adapt an 11B model on a single A6000 (48 GB) GPU using just 20.4 GB of memory. This parameter-efficient strategy makes large-scale MLLM fine-tuning feasible under limited hardware constraints, while still maintaining strong performance. Experimental results indicate that both methods achieve competitive accuracy on multimodal tasks such as medical VQA. Further comparisons between LoRA and QLoRA are discussed in Section 5.

### 3.3. Training Pipeline

As illustrated in Figure 1(b), we propose a two-stage fine-tuning pipeline based on the aforementioned model architecture. The first stage, referred to as biomedical alignment, equips the model with fundamental capabilities in medical image understanding and generation of image descriptions. The second stage, downstream fine-tuning, aims to further enhance the model's performance on medical visual question answering tasks. The overall training procedure involves several key components, including image preprocessing, instruction prompt design, the two-stage fine-tuning process, and parameter-efficient fine-tuning strategies. Detailed descriptions of these components are provided in the following sections.

**Data Preprocessing and Input Formatting**   Before training, all medical images were uniformly preprocessed by resizing them to a resolution of $512 \times 512$. Experiments show that this setting effectively preserves key visual information while significantly reducing the risk of out-of-memory (OOM) errors during training, leading to more stable and efficient model convergence and better model performance.

Since the LLaMA 3.2 base model was pre-trained in an instruction-following chat format, all downstream datasets were reformatted into a compatible conversational structure to enable effective two-stage instruction tuning.

| Biomedical Alignment (Stage 1) | Downstream Finetuning (Stage 2) |
|---|---|
| User: <image> You are an expert radiologist. Describe accurately what you see in this medical image. Assistant: [caption] | User: <image> You are an expert radiologist. Answer the following medical question concisely: [question] Assistant: The answer is: [answer] |

Figure 2: Instruction prompt templates designed for two-stage fine-tuning

**Biomedical Alignment (Stage 1)**   In the first stage of our training pipeline, we perform instruction tuning using the training split of the ROCOv2-Radiology (Rückert et al., 2024) dataset. The dataset contains 60K medical image–caption pairs collected from PubMed Central, spanning a wide range of anatomical regions and medical imaging modalities.

Each sample from the dataset is reformatted into the structure illustrated in Figure 2, where *[caption]* field denotes the ground truth textual description associated with the medical image and serves as the training target.

Unlike prior studies such as Med-Flamingo (Moor et al., 2023), LLaVA-Med (Li et al., 2023a), and HuotuoGPT-Vision (Chen et al., 2024), which typically adopt full-parameter

fine-tuning strategies that update only the vision encoder during the pretraining stage, we employ a parameter-efficient approach. Specifically, we insert LoRA (Hu et al., 2022) adapters into all linear layers of the vision encoder, multimodal projector, and the LLM, and fine-tune only the adapter parameters across all three modules. This unified PEFT strategy enables efficient adaptation while significantly reducing training cost.

**Downstream Fine-Tuning (Stage 2)**   In the second stage, we further fine-tune the biomedical-aligned model on three medical visual question answering datasets: SLAKE (Liu et al., 2021), VQA-RAD (Lau et al., 2018), and PathVQA (He et al., 2020). The model is initialized from the checkpoint obtained from Stage 1. Consistent with the previous stage, we inject LoRA adapters into the fully connected layers of the vision encoder, multimodal projector, and the LLM, and fine-tune only the adapter parameters within these modules.

SLAKE (Liu et al., 2021) is a bilingual medical VQA dataset containing 642 medical images and 7,032 QA pairs. In this study, we use only the English subset. VQA-RAD (Lau et al., 2018) consists of 315 radiology images and 3,515 QA pairs. PathVQA (He et al., 2020) focuses on pathological image understanding and includes 4,998 pathology slide images with 32,795 QA pairs in total. Collectively, these datasets span a wide range of medical questions, including organ identification, lesion localization, abnormality detection, and modality classification, covering both closed-ended and open-ended question formats. They serve as standard benchmarks for evaluating model performance in radiology and pathology visual reasoning tasks.

Each sample in the datasets is reformatted into the structure shown in Figure 2, where the *[question]* and *[answer]* fields are taken from the original dataset annotations. This prompt format aligns with the dialogue-based structure used during model training and facilitates efficient output extraction during inference. During training, the model's tokenizer converts the structured dialogue into a token sequence, which is subsequently mapped to corresponding integer IDs. Only the tokens corresponding to the assistant's response are used to compute the loss, allowing the model to update its parameters accordingly. Following the auto-regressive modeling paradigm, the model learns to predict the next token in the sequence, progressively generating a coherent response.

Through this two-stage PEFT training framework, we effectively adapt a state-of-the-art general-domain multimodal large language model to the medical domain. The resulting system is capable of interpreting medical images, answering clinical queries, and generating accurate diagnostic descriptions and clinically relevant responses, laying a solid foundation for intelligent vision–language applications in healthcare.

**Experimental Details**   Full specifications of the training configurations and hyperparameter settings are included in Appendix B.

## 4. Evaluation and Results

We conducted a comprehensive evaluation of our model on multiple datasets for medical visual understanding, benchmarking its performance on medical VQA datasets against several representative state-of-the-art (SOTA) methods. In addition, we applied our two-stage fine-tuning strategy to adapt general-purpose multimodal models such as LLaVA (Liu et al., 2023), Qwen2.5-VL-7B (Bai et al., 2025) and Gemma3-4B (Team et al., 2025), and

Table 1: Accuracy (%) comparison across SOTA methods, our fine-tuned model, and its zero-shot version on three medical VQA datasets. '–' denotes results not reported in the original paper.

| Methods | Type | SLAKE | | | PathVQA | | | VQA-RAD | | |
|---|---|---|---|---|---|---|---|---|---|---|
| | | Closed | Open | Overall | Closed | Open | Overall | Closed | Open | Overall |
| **Representative SOTA methods** | | | | | | | | | | |
| M2I2 (Li et al., 2023b) | Non-LLMs | 91.1 | 74.7 | 81.2 | 88.0 | 36.3 | 62.2 | 83.5 | 66.5 | 76.8 |
| VQA-Adapter (Liu et al., 2024b) | | 83.7 | 79.2 | 81.0 | – | – | – | 82.3 | 66.1 | 75.8 |
| MMQ (Do et al., 2021) | | – | – | – | 84.0 | 13.4 | 48.8 | 75.8 | 53.7 | 67.0 |
| PubMedCLIP (Eslami et al., 2021) | | 82.5 | 78.4 | 80.1 | – | – | – | 80.0 | 60.1 | 72.1 |
| CLIP-ViT w/GPT2 (Van Sonsbeek et al., 2023) | LLMs | 82.1 | 84.3 | 83.3 | 87.0 | 40.0 | 63.6 | – | – | – |
| BiomedGPT (Zhang et al., 2024) | | 89.9 | 84.3 | 86.1 | 88.0 | 28.0 | 58.1 | 81.3 | 60.9 | 73.2 |
| LLaVA-Med (Li et al., 2023a) | | 83.1 | 79.3 | 80.6 | 89.8 | 32.7 | 61.3 | 81.2 | 55.0 | 69.6 |
| HuatuoGPT-Vision (Chen et al., 2024) | | 76.1 | 59.8 | 65.2 | 62.5 | 20.1 | 41.3 | 74.1 | 43.5 | 60.5 |
| MedGemma (Sellergren et al., 2025) | | 73.2 | 57.2 | 63.9 | 65.1 | 20.4 | 47.7 | 72.9 | 40.5 | 58.5 |
| **Zero-shot result** | | | | | | | | | | |
| LLaMA 3.2 Vision Instruct (Meta AI, 2024) | LLMs | 71.9 | 17.4 | 36.1 | 42.7 | 2.7 | 31.3 | 57.1 | 10.3 | 39.0 |
| **Our two-stage fine-tuning result** | | | | | | | | | | |
| LLaMA32-Med | LLMs | 91.2 | 81.4 | 84.6 | 92.4 | 33.7 | 63.1 | 81.7 | 57.5 | 70.9 |
| LLaMA32-Med (Human Evaluation) | LLMs | 91.2 | 84.0 | 86.4 | 92.4 | 38.5 | 65.5 | 81.7 | 62.5 | 74.2 |

included them in the comparative analysis. Experimental results demonstrate that our model achieves performance comparable to or better than existing SOTA methods and other fine-tuned multimodal models across a range of evaluation metrics, while significantly outperforming its own zero-shot variant.

## 4.1. Performance on Medical VQA Benchmarks

We evaluate our model on the test sets of three medical VQA benchmarks (SLAKE, VQA-RAD, and PathVQA) to assess its performance on medical visual question answering. As shown in Table 1, our two-stage fine-tuned model LLaMA32-Med achieves the highest accuracy on closed-ended questions for both SLAKE and PathVQA. On the PathVQA-Closed subtask, LLaMA32-Med reaches 92.4% accuracy, outperforming BiomedGPT and LLaVA-Med by 4 and 3 percentage points, respectively. It also outperforms recent multimodal medical LLMs such as HuatouGPT-Vision and MedGemma, highlighting the effectiveness of our lightweight PEFT framework.

For overall accuracy, LLaMA32-Med achieves 84.6% on SLAKE and 63.1% on PathVQA, outperforming larger models such as BiomedGPT and matching or exceeding several competitive baselines. Performance on open-ended questions is slightly weaker, which may be related to the limited number of trainable parameters and the modest size of the Stage 1 instruction-tuning corpus. Moreover, LLaMA32-Med yields substantial improvements over its zero-shot counterpart, achieving gains of 48 percentage points on SLAKE and approximately 32 points on both PathVQA and VQA-RAD.

Qualitative examples are shown in Figure 3, which compares LLaMA32-Med predictions with the ground truth in the test set. The examples cover diverse question types, including anatomical localization, modality identification, organ-level reasoning, and disease identification. While the model occasionally struggles with complex disease-related queries, it produces largely correct answers across clinically relevant tasks. These results indicate that

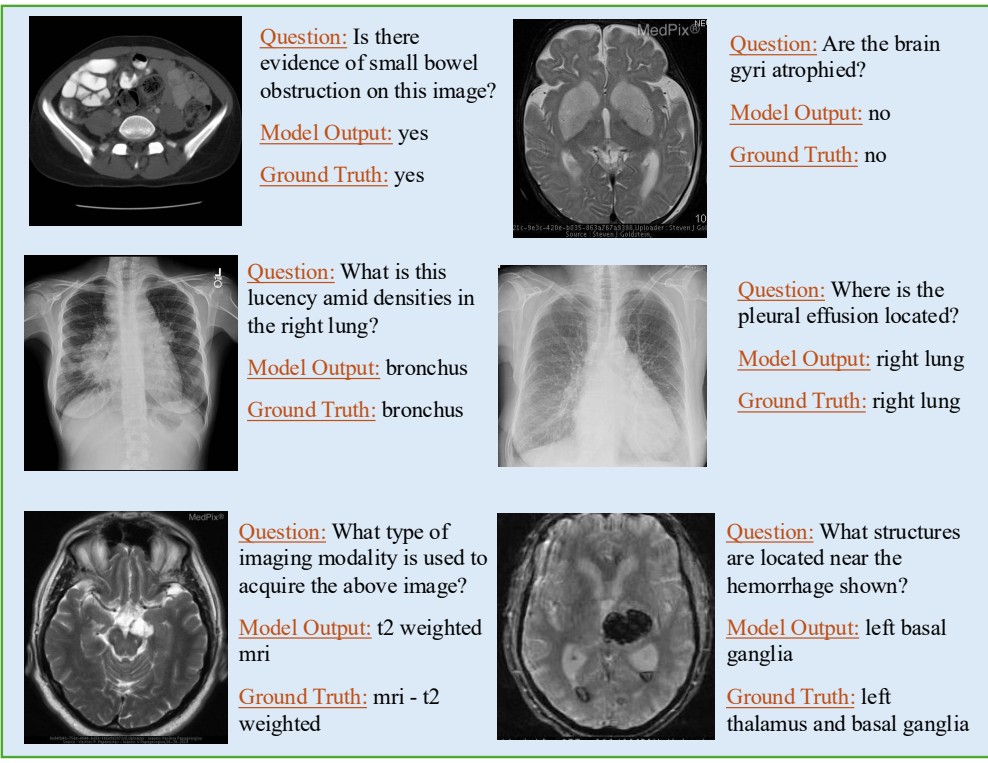

Figure 3: Qualitative medical VQA examples of LLaMA32-Med.

our fine-tuning strategy effectively injects domain-specific medical knowledge and enhances the adaptability of general-domain MLLMs for medical VQA tasks.

**Human evaluation** Traditional string matching often fails to capture true correctness in open-ended questions. For example, "cardiomegaly" and "enlarged heart" are clinically equivalent but are marked incorrect under strict lexical matching. To further evaluate performance on open-ended questions, we invite three clinical experts to conduct human assessment, and we report the averaged score. Under human expert evaluation, LLaMA32-Med shows an improvement of roughly 2-3 percentage points, indicating that human review more accurately reflects the model's clinical understanding than string matching.

## 4.2. Comparative Experiments on Multiple MLLMs

We further compared the proposed LLaMA32-Med model with three fine-tuned advanced multimodal baselines: LLaVA, Qwen2.5-VL, and Gemma3. To ensure fairness, all foundation MLLMs were trained using the same two-stage fine-tuning pipeline and evaluated on three medical VQA benchmark datasets using overall accuracy as the metric. As shown in Figure 4, all four fine-tuned models outperform their zero-shot counterparts by a large margin. This demonstrates that the proposed two-stage fine-tuning pipeline substantially enhances the performance of general-purpose MLLMs on medical vision–language tasks, achieving significant gains in medical VQA while updating less than 2% of parameters.

In addition, as illustrated in Figure 4, LLaMA32-Med achieves an accuracy of 84.6% on the SLAKE dataset, outperforming the fine-tuned LLaVA, Qwen2.5-VL, and Gemma3

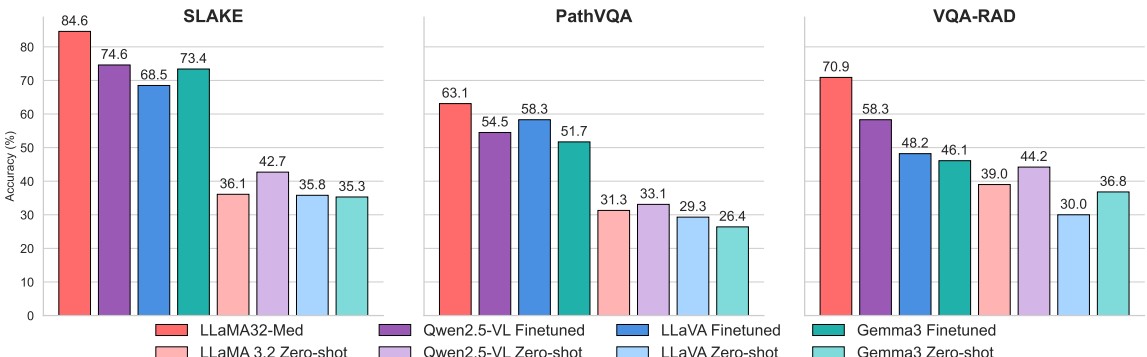

Figure 4: Finetuned and Zero-shot MLLMs Comparison Across Medical VQA Datasets

models. On the VQA-RAD dataset, LLaMA32-Med again achieves the best performance with an accuracy of 70.9%, surpassing Qwen2.5-VL (58.3%), LLaVA (48.2%), and Gemma3 (46.1%). On the PathVQA dataset, LLaMA32-Med once more ranks first among all models. Although its parameter size is slightly larger than the other three models, all models were fine-tuned with minimal parameter update. The significant performance improvements suggest that these gains may be attributed to LLama32-Med's more effective cross-modal attention fusion mechanism and the proposed two-stage PEFT training strategy.

### 4.3. Results for Medical Image Description

**Evaluation Method** To assess our model's performance on image description after biomedical alignment, we randomly sampled 1,000 test instances from the ROCOv2-radiology dataset, and reported the average scores across all samples. Traditional metrics such as BLEU and Recall have known limitations in capturing semantic consistency and clinical relevance (Li et al., 2023a). Therefore, we adopt a semantic evaluation strategy using DeepSeek (Liu et al., 2024a) as an LLM-as-a-judge.

We input the generated descriptions from our model, the original LLaMA 3.2 Vision-Instruct model, and the ground truth captions into the DeepSeek interface, with task-specific prompts. The model outputs scores ranging from 1 to 5, evaluating semantic alignment and clinical accuracy between the generated medical description and the ground truth. The specific prompt template is provided in Appendix C. This approach offers a more reliable and context-aware assessment compared to traditional lexical metrics, particularly for open-ended medical image descriptions.

**Quantitative Results** While DeepSeek-based score serves as our primary evaluation metric, we also report BLEU, Recall and BERTScore for completeness. As shown in Table 2, the BLEU score of the fine-tuned model is 0.047, nearly double that of the original model (0.026), with improvements in Recall and BERTScore as well. These metrics, however, remain limited for evaluating open-ended generation tasks. In contrast, the DeepSeek-based semantic score shows a clear improvement: 3.80 for the fine-tuned model versus 2.56 for the original, indicating enhanced medical image understanding and clinical relevance enabled by biomedical alignment.

Table 2: Average performance comparison between our LLaMA32-Med and the original LLaMA 3.2 on medical image captioning, evaluated on 1000 samples from ROCOv2-Radiology test set.

| Model | Deepseek-based score (1–5) | BLEU | Recall | BERTScore |
|---|---|---|---|---|
| LLaMA32-Med | 3.80 | 0.047 | 0.358 | 0.8467 |
| Original LLaMA 3.2 | 2.56 | 0.026 | 0.275 | 0.8165 |

**Qualitative Results**  To illustrate the impact of biomedical alignment fine-tuning, we present two representative examples from the ROCO dataset. In Figure 5, we observe that the fine-tuned LLaMA32-Med generates more precise and lesion-focused descriptions that closely align with the ground truth, such as correctly identifying fractures or detecting tumors and specifying their anatomical position. In contrast, the original model tends to produce irrelevant anatomical details and struggles to determine specific disease entities. Across other samples, the original model also frequently hallucinates diagnoses or patient history, whereas the fine-tuned model remains consistent with the image content. Overall, biomedical alignment fine-tuning strengthens the model's medical visual understanding and clinical reasoning capabilities, providing a solid foundation for adapting general-domain models into domain-specific medical MLLMs. These observations further validate the effectiveness of our proposed two-stage fine-tuning method.

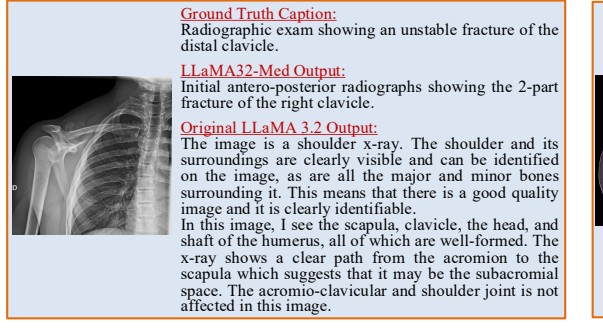 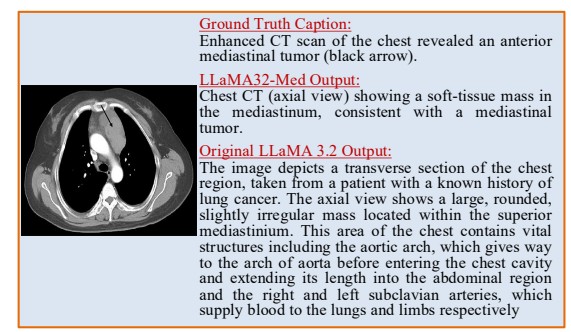

Figure 5: Example comparison of medical image description generated by LLaMA32-Med, original LLaMA 3.2, and the ground truth.

## 5. Discussion

Table 3: Ablation study of different fine-tuning strategies evaluated on the SLAKE dataset.

| Method | Stage 1 FT | Role-based Prompt | Vision Encoder FT | 4-bit Quantization | Accuracy (%) |
|---|---|---|---|---|---|
| LoRA (full) | ✓ | ✓ | ✓ | | 84.6 |
| LoRA w/o stage 1 FT | | ✓ | ✓ | | 79.1 |
| LoRA w/o role-based prompt | ✓ | | ✓ | | 82.4 |
| LoRA w/o vision encoder FT | ✓ | ✓ | | | 80.4 |
| QLoRA (full) | ✓ | ✓ | ✓ | ✓ | 81.8 |

**Ablation Study**  We conducted an ablation study on SLAKE test set to assess the contribution of each component in our fine-tuning pipeline. As shown in Table 3, the best

performance is achieved when Stage 1 biomedical alignment, vision encoder fine-tuning, and radiologist role-based prompting are all applied together. Removing Stage 1 leads to an approximate 5% accuracy drop, while skipping vision encoder fine-tuning results in about 4% degradation, highlighting the importance of adapting both the language and visual components to medical data. Role-based prompting further provides a 2.2% improvement over a generic template, confirming the benefit of domain-specific instructions. Additionally, LoRA outperforms QLoRA by roughly 3%, suggesting that although QLoRA offers memory efficiency, it comes with a modest trade-off in accuracy. Overall, each component contributes meaningfully, and the full configuration yields the most robust results.

**Limitations**  Despite the strong performance of our approach on the medical VQA task, some limitations remain. We have not explored long-form, structured medical report generation, which falls under a different research direction and requires substantial report-level data for fine-tuning. This aspect will be considered for our future work. In addition, clinical experts are involved in reviewing cases where string matching fails, which may become inefficient for large datasets. Future studies may incorporate LLM-as-a-judge mechanisms to more effectively assess the model's accuracy relative to the ground truth.

## 6. Conclusion

In this study, we present a parameter-efficient fine-tuning (PEFT) framework to adapt the general-domain LLaMA 3.2 Vision-Instruct model for medical image–text understanding. Our two-stage training pipeline with task-specific and role-based prompts enhances medical visual understanding and question answering ability while keeping computational cost low.

Our experiments show that the LLaMA32-Med achieves performance comparable to or exceeding several SOTA methods and substantially outperforms the original general-domain version, demonstrating strong adaptability for real-world medical applications. Overall, these results highlight the effectiveness of our PEFT-based approach in improving multimodal medical understanding, and lay the groundwork for building efficient, clinically reliable decision-support systems to advance personalized medicine in the future.

## Acknowledgements

This research is supported by A*STAR, CISCO Systems (USA) Pte. Ltd and National University of Singapore under its Cisco-NUS Accelerated Digital Economy Corporate Laboratory (Award I21001E0002). We also thank the Kent-Ridge AI research group at the National University of Singapore for helpful discussions.

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

## Appendix A. Model Architecture Details

As illustrated in Figure 1(a), LLaMA 3.2 Vision-Instruct model comprises three main components: a vision encoder, a multimodal projector, and a language model. The vision encoder adopts a two-stage hierarchical Transformer architecture, which represents one of the core innovations of the model. In the first stage, the local vision encoder performs patch embedding on the input image and extracts local visual features while retaining several intermediate hidden states. These features are then passed to the global vision encoder for high-level contextual modeling. The final visual representation is obtained by concatenating all intermediate features from the local encoders with the global output, forming a unified visual embedding. The multimodal projector, implemented as a multilayer perceptron (MLP), maps the joint visual embedding into the semantic space of the language model, aligning the visual and textual modalities. The projected visual tokens are then used as the input to the language model.

The language model is based on the decoder-only Transformer architecture from LLaMA 3.1, with cross-attention layers inserted every five Transformer blocks. In this mechanism, text tokens encoded by the LLM act as queries, while the projected visual features serve as keys and values in attention modules. This enables periodic incorporation of visual context throughout the generation process. The interleaved attention structure allows the model to continuously integrate visual semantics into the text generation, significantly enhancing its vision-language understanding and generative capabilities.

## Appendix B. Experiment Settings

The training was conducted in a Python 3.11 and CUDA 12.4 environment using one NVIDIA A6000 GPUs. Input images were uniformly resized to a resolution of 512×512, and the maximum input text length was set to 1024 tokens.

The model was initialized with pretrained weights from LLaMA 3.2 Vision-Instruct in the first stage. Parameter-Efficient Fine-Tuning (PEFT) strategy was applied across both training stages, with the LoRA adapter rank set to 32. In the first stage (biomedical alignment), the model was trained for 3 epochs with an initial learning rate of 2e-4. In the second stage (downstream fine-tuning), training was conducted for 3 epochs with an initial learning rate of 4e-5. The AdamW optimizer was used with a weight decay of 0.01,

and a cosine learning rate scheduler was employed to dynamically adjust the learning rate throughout the training process.

## Appendix C. DeepSeek Prompt Template

Figure 6 presents the evaluation prompt we used to obtain the DeepSeek-based score (1–5 scale) for assessing semantic consistency and clinical correctness.

> You are an expert radiologist, assess the following generated radiology report description by comparing it to the provided ground truth caption.
>
> Evaluate both the semantic similarity and clinical accuracy between the two  Assign a score on a scale from 1 to 5, where 1 indicates no similarity or accuracy, and 5 signifies high similarity and accuracy.
>
> Ensure that higher scores are given when the generated description closely matches the ground truth in both meaning and clinical correctness.
>
> Ground truth caption: {ground_truth}
>
> Model generated description: {generated_text}
>
> Format your response as:
>
> Reasoning: [Your reasoning]
>
> Score: X

Figure 6: Prompt template used for the DeepSeek-based evaluation.

