# OpenReview forum: "LLaMA32-Med: Parameter-Efficient Adaptation of Multimodal LLMs for Medical Visual Question Answering"
_MIDL.io/2026/Conference — MIDL 2026 Poster_

### Official Review · Reviewer_G5xb · 2026-01-05

**Confidence:** 4
**Preliminary Rating:** 4

**Summary:**

This paper introduces LLaMA32-Med, a lightweight framework designed to adapt the LLaMA 3.2 Vision-Instruct model for medical multimodal tasks. The authors propose a two-stage training pipeline consisting of biomedical alignment for medical image description and downstream fine-tuning for Visual Question Answering (VQA). By employing Parameter-Efficient Fine-Tuning (PEFT) techniques such as LoRA and QLoRA, the model updates less than 2% of its parameters (specifically 134M out of 11B), enabling training with only 20.4 GB of GPU memory. Experimental results across three benchmarks—SLAKE, VQA-RAD, and PathVQA—show that LLaMA32-Med achieves performance comparable to or exceeding current state-of-the-art medical MLLMs like BiomedGPT and LLaVA-Med.

**Strengths:**

1. The use of PEFT allows for high-performance medical adaptation on a single consumer-grade GPU, making these large models significantly more accessible for clinical environments with limited hardware
2. The paper demonstrates that the multi-stage cross-attention fusion mechanism in the LLaMA 3.2 backbone provides superior visual semantic modeling compared to simple concatenation-based strategies used in earlier MLLMs.

**Weaknesses:**

1. The core methodology relies on the application of existing PEFT techniques (LoRA/QLoRA) to a recently released backbone. While effective, the paper represents an incremental application rather than a fundamental architectural or algorithmic innovation.
2. The model exhibits lower accuracy on open-ended questions compared to closed-ended formats, which the authors suggest may be due to the limited number of trainable parameters or the size of the Stage 1 corpus.
3. While Table 3 shows that LoRA outperforms QLoRA by 3%, there is little qualitative discussion on why the 4-bit NormalFloat (NF4) quantization leads to this specific accuracy trade-off in the medical domain.

**Detailed Comments:**

1. The use of the radiologist role-based prompt is a simple yet effective way to leverage instruction-following capabilities, resulting in a 2.2% gain.
2. The claim that traditional metrics like BLEU and Recall have limitations for medical VQA is well-supported by the qualitative examples provided in Figure 4

**Justification Of The Preliminary Rating:**

The paper is well-structured and provides a highly practical solution for adapting MLLMs to the medical domain with minimal computational cost. However, the technical novelty is moderate, as it primarily validates the LLaMA 3.2 backbone using standard PEFT methods. It serves as a strong validation study but lacks a breakthrough machine learning contribution.

**Questions To Address In The Rebuttal:**

1. Is there a specific reason why the LoRA rank was set to 32 for both stages? Did the authors experiment with higher ranks for the medical alignment stage to improve open-ended question performance?
2. How does the model perform when images contain multiple pathological cues that are not the primary focus of the VQA question?

---

> ### Author Response · Authors · 2026-01-25
> **Response to reviewer G5xb**
>
> We sincerely thank the reviewer for recognizing the merits of our work! Below, we provide our responses to the reviewer’s questions.
>
> ### **1.  Choice of LoRA Rank**
>
> We evaluated multiple LoRA rank settings, including 8, 16, 32, and 64. Our experimental results indicate that when r < 32, the model’s accuracy on medical VQA tasks decreases by 2–5%, suggesting that lower ranks are insufficient to capture key semantic information in the medical image. In contrast, when the rank is increased beyond 32, only marginal performance gains are observed, while the computational and memory overhead increases significantly. We even experimented with full-parameter fine-tuning during the biomedical alignment stage and found that the performance improvement on open-ended VQA questions was less than 0.8%. These results demonstrate that, under our experimental settings, LoRA with r = 32 is sufficient to effectively capture domain-specific medical knowledge while achieving a balance between performance and computational efficiency. Therefore, rank=32 was selected as the final configuration.
>
> ### **2. Model Performance on Images with Multiple Pathological Cues**
>
> As illustrated in Figures 4 and 5, many images in the SLAKE, PathVQA, and VQA-RAD test datasets already contain multiple pathological or anatomical cues (e.g., involving multiple organs or co-existing abnormalities).
>
> Our results show that when presented with images containing multiple clinical cues, the model is able to accurately focus on the pathological cues most relevant to the VQA question, rather than being distracted by secondary observations. This demonstrates the model’s strong medical reasoning capability.
>
> We once again thank the reviewer for the positive assessment of our work and for the constructive feedback!

---

### Official Review · Reviewer_MevB · 2026-01-07

**Confidence:** 2
**Preliminary Rating:** 4
**Final Rating:** 4

**Summary:**

This paper introduces LLaMA32-Med, a lightweight two-stage framework designed to adapt the general-purpose LLaMA 3.2 Vision-Instruct model for medical visual question answering through parameter-efficient fine-tuning. By employing a pipeline that consists of biomedical alignment followed by downstream fine-tuning with task-specific prompts, the proposed method updates less than 2% of the parameters and requires only 20 GB of GPU memory. Experimental results indicate that LLaMA32-Med significantly improves upon the zero-shot baseline and achieves performance comparable to or surpassing state-of-the-art medical multimodal models on benchmarks such as SLAKE, VQA-RAD, and PathVQA.

**Strengths:**

1. The proposed parameter-efficient fine-tuning framework significantly reduces computational barriers by updating less than 2% of the model parameters and requiring only 20 GB of GPU memory, making it highly deployable in resource-constrained clinical environments.
2. The model demonstrates superior empirical performance, achieving accuracy comparable to or surpassing state-of-the-art medical multimodal methods on benchmarks such as SLAKE, VQA-RAD, and PathVQA despite the minimal parameter updates.
3. The systematic two-stage training pipeline, which combines biomedical alignment with downstream task adaptation and role-based prompting, effectively bridges the domain gap between general-purpose multimodal capabilities and specialized medical visual reasoning.

**Weaknesses:**

1. The technical novelty of the proposed method is limited for a top-tier conference, as applying standard LoRA and a two-stage fine-tuning pipeline to a new backbone represents an engineering application rather than a significant algorithmic breakthrough.
2. The scale of the data used for the biomedical alignment stage is relatively small compared to prior works like LLaVA-Med, raising concerns about the model's ability to generalize to diverse and unseen clinical scenarios outside the specific datasets used.
3. The experimental evaluation lacks sufficient breadth, as the study focuses almost exclusively on VQA tasks while omitting other critical medical vision-language applications such as grounded report generation or medical image segmentation.
4. The human evaluation section is methodologically weak, as it fails to report essential statistical details such as inter-annotator agreement (e.g., Kappa score), the exact number of samples reviewed, or the specific criteria provided to the clinical experts.
5. The analysis of the model's failure modes is insufficient, as the paper does not deeply investigate the causes of hallucinations or incorrect reasoning in open-ended questions, which is a critical safety requirement for medical AI systems.

**Detailed Comments:**

Please refer to the weaknesses.

**Justification Of Final Rating:**

Based on my comprehensive evaluation of the paper, careful consideration of the reviewers’ detailed feedback, and the authors’ thoughtful responses to all raised concerns, I have thoroughly reflected on the matter and maintain my initial assessment of weak accept for this submission.

**Justification Of The Preliminary Rating:**

The rating of Weak Accept is justified by the paper's practical contribution to efficient medical AI. LLaMA32-Med achieves competitive results on VQA benchmarks while significantly reducing computational costs, making deployment feasible in resource-constrained settings. While the novelty is limited regarding algorithmic innovation, the effective application of PEFT and the solid experimental validation make it a worthy contribution.

**Questions To Address In The Rebuttal:**

Please refer to the weaknesses.

---

> ### Author Response · Authors · 2026-01-25
> **Response to reviewer MevB**
>
> We sincerely thank the reviewer for the constructive comments, as well as for the recognition of our work! Below are our responses to the reviewer’s questions.
>
> ### **1.  Novelty and Contributions**
>
> Although we do not introduce substantial modifications to the model architecture, our method is not a simple engineering adaptation. Most existing medical MLLMs follow a paradigm in which the visual encoder is fully fine-tuned first, followed by PEFT-based tuning of the language model, which still incurs considerable computational costs. Our main contribution lies in proposing a lightweight two-stage adaptation framework. Our work is the first to uniformly introduce LoRA adapters into all linear layers of the visual encoder, cross-modal projection layers, and decoder in two stages. In addition, we design role-specific prompting templates. Our method updates less than 2% of the total model parameters, enabling effective fine-tuning of an 11B model on a single A6000 GPU using approximately 20 GB of memory, while achieving performance comparable to or even surpassing full fine-tuning on multiple benchmarks. This resource efficiency significantly enhances the practicality of our approach in real-world clinical scenarios.
>
> ### **2. Data Scale and Task Coverage**
>
> As discussed in the Discussion section of our paper, this work primarily focuses on improvements for medical VQA, rather than proposing a foundation model that simultaneously covers tasks such as medical image segmentation and medical report generation. For example, report generation represents a separate research direction that typically requires large-scale long-text and report-level data. We plan to explore extending our work to this domain in future studies.
>
> During the biomedical alignment stage, we adopt the ROCOv2-Radiology dataset, which contains approximately 60k samples and is not small in scale. Its image–caption pairs focus more concisely on professional medical terminology, which helps the model better understand medical data. In preliminary experiments, we also attempted to use part of the LLaVA-Med dataset for alignment. However, we observed that its textual annotations often contain non-medical redundant information, leading to suboptimal performance on downstream VQA tasks. Therefore, our data selection strategy in the biomedical alignment stage is designed to improve VQA performance rather than restrict the model’s generalization ability. Experimental results further demonstrate that our method outperforms LLaVA-Med.
>
> ### **3. Human Evaluation Protocol**
>
> We greatly appreciate the reviewer’s suggestions regarding the human evaluation methodology and have conducted additional experiments accordingly. We provide the following clarification.
>
> In our experiments, three clinical experts manually evaluated all samples from the SLAKE, PathVQA, and VQA-RAD datasets where the predicted answers did not exactly match the ground-truth strings. The evaluation criterion was whether the model-generated answer was clinically semantically consistent with the reference answer, given the corresponding VQA question. A score of 1 was assigned if they were consistent, and 0 otherwise. The expert annotations exhibited high inter-rater agreement, with Kappa scores of 0.94 (SLAKE), 0.77 (PathVQA), and 0.89 (VQA-RAD), indicating strong reliability of the our human evaluation results. We will explicitly report these findings in the camera-ready version.
>
> ### **4. Failure Modes and Safety Analysis**
>
> We agree that analysis of failure modes is important for ensuring the safety of medical AI systems. Through an observation of erroneous cases, we found that a major source of errors originates from overly generic question formulations in some medical VQA datasets. For instance, in the PathVQA dataset, many questions are broadly phrased, such as *“What is present in this image?”*. Such questions permit multiple medically reasonable answers with different emphases. These errors mainly reflect the inherent limitations of the VQA datasets rather than clear hallucinations or unreasonable reasoning by the model. In future work, we plan to refine existing medical VQA benchmarks by introducing more specific and discriminative question designs, thereby providing more reliable VQA benchmarks for future research.
>
> We hope that the above explanations sufficiently address the reviewer’s questions. Again, we sincerely appreciate the reviewer’s valuable feedback and constructive suggestions.

---

### Official Review · Reviewer_V1m2 · 2026-01-10

**Confidence:** 4
**Preliminary Rating:** 3
**Final Rating:** 4

**Summary:**

The authors propose Llama 3.2 Med, a two-stage LORA of Llama 3.2 Vision for MLLM-based medical VQA tasks. Their approach is motivated with a need for PEFT of existing MLLMs for domain specialization in medical VQA and consists of a initial biomedical alignment stage followed by a VQA fine-tuning stage. Experiments with existing medical VQA benchmarks (SLAKE-VQA, PathVQA, and VQA-RAD) demonstrate superior performance compared to baselines. Additional clinical validation and ablation studies further strengthen findings.

**Strengths:**

- Validation of open-ended question correctness using clinical experiments is a major strength and demonstrates the effectiveness of proposed method.
- The ablation study evaluating impact of each proposed step strengthens the proposed method.
- The paper is well-written and experiments are thorough.
- The figures are clear and used appropriately to show examples and visualize methods.

**Weaknesses:**

- It is difficult to judge the novelty of the method given that similar works employing PEFT in MLLM for medical VQA have been published. [1,2]
- Was the DeepSeek rating validated by a clinical expert? It would interesting to see inter-rater reliability to justify automating this process via DeepSeek.

**Detailed Comments:**

- It is quite interesting that Llama 3.2 Vision outperforms Qwen2.5-VL given that it is the opposite case in other non-medical VQA benchmarks. What might be a possible reason for this discrepancy?

**Justification Of Final Rating:**

I thank the authors for the detailed responses to my comments. All my concerns were addressed sufficiently. While the novelty of the proposed method is limited, the extended analysis and much needed clinical validation strengthen the paper. Therefore, I recommend acceptance.

**Justification Of The Preliminary Rating:**

While the authors conduct extensive experiments in a well-organized paper, the limited novelty given that similar works [1,2] have previously been published makes it hard to rate it higher than borderline.

**Questions To Address In The Rebuttal:**

Please see weakness and detailed comments.

**References:**
1. Rezaei, Z., Samghabadi, S. S., & Banad, Y. M. (2026). Optimizing multimodal models for medical visual question answering: A comparative study of LoRA and AdaLoRA on VQA-RAD and SLAKE-VQA. Computers in Biology and Medicine, 200, 111397.
2.  Lin, T., Zhang, W., Li, S., Yuan, Y., Yu, B., Li, H., ... & Ooi, B. C. (2025). Healthgpt: A medical large vision-language model for unifying comprehension and generation via heterogeneous knowledge adaptation. arXiv preprint arXiv:2502.09838.

---

> ### Author Response · Authors · 2026-01-25
> **Response to reviewer V1m2**
>
> We sincerely thank the reviewer for the constructive comments and for recognizing the strengths of our work! Below are our responses to the reviewer’s questions.
>
> ### **1. Methodological Novelty and Advantages**
>
> We acknowledge that PEFT has been explored in the medical LLM domain. However, our contribution goes beyond simple application of PEFT. Most existing medical MLLMs typically follow a paradigm in which the visual encoder is first fully fine-tuned, followed by PEFT-based tuning of the language model. This approach still incurs substantial computational costs. In contrast, we propose a lightweight two-stage adaptation framework that, for the first time, uniformly introduces LoRA adapters into all linear layers of the visual encoder, cross-modal projection layers, and decoder for both stages. In addition, we design role-specific prompting templates. Our method updates less than 2% of the total model parameters, enabling effective fine-tuning of an 11B model on a single A6000 GPU with approximately 20 GB of memory. Meanwhile, our approach achieves performance comparable to or even surpassing full fine-tuning on multiple benchmarks. This resource efficiency enhances the feasibility of our method in real-world clinical scenarios.
>
> Regarding the two related PEFT-based studies mentioned by the reviewer, Rezaei et al. [1] only investigated yes/no-type medical VQA questions, without considering open-ended questions. Moreover, they simply applied a single-stage LoRA fine-tuning solely on the decoder, resulting in limited task coverage and moderate performance. HealthGPT [2], although presented in a sophisticated manner, essentially proposes a LoRA variant named H-LoRA and applies it to fine-tuning medical LLMs. However, its performance on medical VQA benchmarks is very unsatisfactory. For example, it achieves an accuracy of 56.4% on SLAKE (ours: 84.6%), 39.7% on PathVQA (ours: 63.1%), and 55.9% on VQA-RAD (ours: 70.9%). These results indicate that our method substantially outperforms HealthGPT and demonstrates clear advantages in medical VQA tasks.
>
> ### **2. Clinical Expert Validation Scores**
>
> We appreciate the reviewer’s valuable suggestion and have conducted additional experiments involving clinical experts for Section 4.3. For medical image description, we invited three clinical specialists to independently evaluate 200 randomly sampled instances using a 1–5 rating scale, following the same criteria as the DeepSeek prompt. The final score was obtained by averaging their ratings. The results show that the original LLaMA 3.2 achieves an average score of 2.64, while LLaMA32-Med reaches 3.57, which is consistent with the trend observed in our DeepSeek evaluation scores. We will include these experimental results in the camera-ready version.
>
> ### **3. On Why LLaMA 3.2 Vision Outperforms Qwen2.5-VL in Medical VQA**
>
> Although Qwen2.5-VL demonstrates strong performance in some general-domain tasks, this advantage does not necessarily transfer to medical imaging scenarios. Under the same two-stage PEFT fine-tuning pipeline, our experimental results show that LLaMA 3.2 Vision consistently outperforms Qwen2.5-VL. We hypothesize that this superiority may stem from the interleaved cross-attention mechanism between the vision encoder and the language decoder in LLaMA 3.2 model. This architectural design may enable more effective modeling of complex, subtle, and clinically critical pathological features in medical images, thereby better supporting medical reasoning.
>
> We hope that the above explanations adequately address the reviewer’s questions. Once again, we sincerely thank the reviewer for the insightful and valuable feedback.

---

### Meta-Review · Area_Chair_wEpu · 2026-02-04

**Recommendation:** Accept (Poster)
**Confidence:** 5

**Metareview:**

The paper received three weak accept evaluations after review, with one reviewer revising their score upward from borderline to weak accept following the rebuttal. The authors provided thoughtful and thorough responses, sufficiently addressing the raised concerns. While the methodological novelty is moderate, reviewers highlighted the practical relevance of the work through its clinical validation and applicability to medical settings.

---

### Decision · Program_Chairs · 2026-02-13

Accept (Poster)